# Neoadjuvant Therapy in Pancreatic Ductal Adenocarcinoma: Aligning Guideline Recommendations with Real-World Evidence

**DOI:** 10.3390/cancers17183085

**Published:** 2025-09-22

**Authors:** Roberto Cammarata, Alberto Catamerò, Vincenzo La Vaccara, Roberto Coppola, Damiano Caputo

**Affiliations:** 1Operative Research Unit of General Surgery, Fondazione Policlinico Universitario Campus Bio-Medico, 00128 Rome, Italy; roberto.cammarata@policlinicocampus.it (R.C.); v.lavaccara@policlinicocampus.it (V.L.V.); r.coppola@policlinicocampus.it (R.C.); d.caputo@policlinicocampus.it (D.C.); 2Research Unit of General Surgery, Università Campus Bio-Medico di Roma, 00128 Rome, Italy

**Keywords:** pancreatic ductal adenocarcinoma, neoadjuvant therapy, borderline resectable, resectable pancreatic cancer, FOLFIRINOX, gemcitabine, chemoradiotherapy, R0 resection, circulating tumor DNA, precision oncology

## Abstract

Pancreatic ductal adenocarcinoma (PDAC) remains one of the most lethal malignancies, with limited long-term survival even after radical surgery. The historical “surgery-first” approach is challenged by two consistent observations: the frequent presence of micrometastatic disease at diagnosis and the difficulty of completing postoperative treatment due to surgical complications or early relapse. Neoadjuvant therapy, delivered before surgery, provides systemic control at an earlier stage, allows biological selection of patients who will benefit from resection, and increases the probability of achieving complete tumor clearance. Randomized trials in borderline resectable disease have demonstrated improvements in surgical margins and survival, while evidence in resectable disease is more heterogeneous but increasingly supportive. Real-world series confirm feasibility and safety, though with substantial variability in regimens and sequencing. Future perspectives focus on integrating molecular profiling, circulating tumor DNA dynamics, and advanced imaging to refine patient selection and guide personalized multimodal strategies.

## 1. Introduction: Challenging the “Surgery-First” Paradigm

Pancreatic ductal adenocarcinoma (PDAC) accounts for more than 90% of exocrine pancreatic malignancies and, according to epidemiological projections, is expected to become the second leading cause of cancer-related mortality by 2040 [1,2]. Despite advances in surgical technique and systemic therapies, prognosis remains dismal, with a 5-year overall survival (OS) rate below 12% [3]. Only 15–20% of patients present with anatomically resectable disease at diagnosis [4], and even in this subgroup recurrence rates exceed 70% after R0 resection followed by adjuvant chemotherapy [5].

Two consistent observations have increasingly challenged the traditional “surgery-first” paradigm. The first is the near-universal presence of micrometastatic disease even in apparently localized tumors [6]. The second is that up to half of patients never receive planned adjuvant therapy due to postoperative complications or early disease progression [7]. These realities have created fertile ground for the development of neoadjuvant therapy (NAT) strategies not only in borderline resectable tumors, where NAT is now endorsed by major guidelines, but also in selected anatomically resectable PDAC.

## 2. Biological and Clinical Rationale for NAT

The rationale for incorporating NAT into the management of PDAC is grounded in both biological principles and pragmatic clinical considerations [8]. From a biological standpoint, PDAC is characterized by an early propensity for systemic spread; autopsy series and molecular studies have shown that micrometastatic deposits can be present at the time of diagnosis even in patients whose disease appears localized on imaging [9]. Administering systemic therapy before surgery provides an opportunity to target these occult lesions at a stage when the tumor burden is lower and the likelihood of chemoresistance may be reduced [10].

Clinically, preoperative delivery of chemotherapy may increase the likelihood that patients receive intended systemic treatment while still in a better physiological state, avoiding the attrition seen with postoperative regimens where surgical complications, delayed recovery, or rapid progression often prevent the completion of adjuvant therapy [10]. This sequencing also acts as a biological filter: patients whose disease remains stable or regresses are more likely to derive benefit from surgery, whereas those who progress are spared the morbidity of an extensive pancreatic resection that would not alter their prognosis [11].

There are additional surgical benefits. NAT can induce tumor shrinkage or fibrosis, facilitating dissection and increasing the likelihood of achieving an R0 resection, an independent predictor of long-term survival [12]. Several studies have reported lower rates of pathological lymph node involvement after NAT, which may translate into reduced locoregional recurrence risk. Moreover, some reports suggest a potential reduction in postoperative pancreatic fistula rates, possibly due to therapy-induced changes in pancreatic texture, although this remains to be confirmed in prospective trials [13].

Historically, radiotherapy was often incorporated into neoadjuvant regimens with the aim of enhancing local control. Retrospective analyses indicate that chemo-radiotherapy may achieve better locoregional disease control compared with chemotherapy alone, particularly in borderline resectable tumors where vessel involvement makes complete clearance challenging [13]. However, the survival benefit of adding radiotherapy remains inconsistent across studies, and its role in the era of highly active systemic regimens such as FOLFIRINOX or gemcitabine/nab-paclitaxel is under active investigation.

Finally, NAT allows for an extended preoperative period during which multidisciplinary teams can perform comprehensive patient optimization, addressing nutrition, biliary drainage, comorbidity control, and psychosocial preparation, all of which may improve perioperative outcomes. Taken together, these factors illustrate why NAT is increasingly viewed not merely as a treatment option, but as a strategic platform for integrating systemic, local, and supportive care interventions in PDAC.

## 3. Evidence from Borderline Resectable PDAC

In borderline resectable PDAC, the evidence supporting a neoadjuvant approach has become increasingly robust over the past decade. Multiple randomized and prospective studies have demonstrated that delivering systemic therapy, often in combination with radiotherapy—before surgery can increase the likelihood of achieving an R0 resection, improve pathological staging, and potentially translate into longer survival.

Among the pivotal trials, the Dutch PREOPANC-1 randomized controlled trial enrolled both resectable (54%) and borderline resectable (46%) patients, comparing gemcitabine-based chemoradiotherapy (36 Gy) followed by surgery with an upfront surgery strategy, both arms receiving adjuvant gemcitabine [14]. While the overall OS difference in the entire cohort was modest (15.7 vs. 14.3 months), NAT significantly improved the R0 resection rate (71% vs. 40%) and disease-free survival, with the magnitude of benefit particularly notable in the borderline subgroup.

The ESPAC-5F trial adopted a more pragmatic design, randomizing patients with borderline resectable PDAC to four different neoadjuvant options—including FOLFIRINOX, gemcitabine/capecitabine (GemCap), and short-course chemoradiotherapy, versus upfront surgery [15]. Despite its relatively small size, the trial reported a striking one-year OS advantage for patients receiving any form of NAT (77%) compared with immediate surgery (39%), corresponding to a hazard ratio of 0.29. Importantly, these results were achieved without a significant increase in perioperative morbidity, underscoring the feasibility of preoperative treatment in this higher-risk population.

The Alliance A021101 pilot study provided additional support for intensive combination therapy in this setting, testing a sequence of mFOLFIRINOX followed by capecitabine-based chemoradiotherapy in borderline resectable patients [16]. Of the patients who proceeded to resection, 93% achieved margin-negative status, and two patients achieved a pathological complete response, a rare event in PDAC. Although this was not a randomized trial and included only 22 patients, it demonstrated the logistical feasibility and potential oncologic benefit of aggressive, multimodality NAT.

Across different regimens and study designs, a consistent pattern has emerged suggesting that NAT may improve R0 resection rates and early survival outcomes in borderline resectable PDAC compared with surgery-first strategies. However, differences in inclusion criteria, chemotherapy backbones, and the use of radiotherapy limit direct cross-trial comparisons.

## 4. Evidence from Resectable PDAC

In anatomically resectable PDAC, the case for neoadjuvant therapy is more nuanced and the evidence base considerably more heterogeneous than in borderline disease. While the theoretical advantages of NAT, early micrometastatic control, improved completion of multimodal therapy, and biological selection-apply equally, the potential benefit must be weighed against the risk of delaying surgery in patients who might already have a curative opportunity.

The Prep-02/JSAP-05 trial from Japan is one of the few phases III randomized studies conducted exclusively in a predominantly resectable population. This multicenter trial compared two cycles of gemcitabine plus S-1 followed by surgery with upfront surgery, both arms receiving adjuvant S-1 [17]. NAT was associated with a clear OS advantage (median 36.7 vs. 26.6 months; HR 0.72), without excess perioperative mortality. This study provided high-level evidence that a short-course neoadjuvant regimen could yield survival gains even in anatomically resectable PDAC, particularly in a population where S-1 is widely used and well tolerated.

In contrast, the SWOG S1505 trial adopted a perioperative approach, randomizing patients with resectable PDAC to either mFOLFIRINOX or gemcitabine/nab-paclitaxel for 3 months before and after surgery [18]. While the median OS was approximately 23 months in both arms, not substantially different from historical surgery-first series—the trial was not powered for direct comparison with an upfront surgery control. Nevertheless, S1505 demonstrated that complex multi-agent chemotherapy can be delivered preoperatively in a high proportion of resectable patients, offering proof-of-concept for NAT feasibility in this setting.

More recently, the NORPACT-1 phase II randomized trial directly compared two months of neoadjuvant mFOLFIRINOX followed by surgery with immediate surgery in resectable PDAC [19]. The study found no OS superiority for the NAT arm and reported a lower rate of patients reaching surgery after randomization, highlighting a key practical challenge: attrition between diagnosis and resection. This drop-off underscores the importance of patient selection and treatment coordination if NAT is to be successfully implemented in resectable disease.

Taken together, and as summarized with relevant studies also for borderline PDAC in Table 1, these studies show that while NAT can be delivered safely and may improve outcomes in certain contexts, the survival advantage in resectable PDAC is not as consistently demonstrated as in borderline disease. Differences in chemotherapy backbone, duration, and trial design make direct comparison difficult, and ongoing phase III trials will be crucial to defining the true role of NAT in this population. At the same time, it should be emphasized that attrition before surgery remains a major limitation of the neoadjuvant approach: in randomized trials, only 60–75% of patients ultimately underwent resection (61% in PREOPANC-1 [14], ~77% in SWOG S1505 [18], and 60% in NORPACT-1 [19]), and less than half completed the entire multimodality pathway including postoperative therapy. These findings highlight that, despite the potential oncologic advantages of NAT, curative-intent surgery remains the most eligible and decisive therapeutic step in the management of resectable PDAC.

## 5. Real-World Evidence: Heterogeneity Behind the Numbers

Beyond the controlled environment of randomized clinical trials, multicenter observational studies from high-volume institutions provide an essential perspective on the actual implementation NAT. Large-scale analyses have confirmed that the advantages seen in experimental settings can be reproduced in routine practice [20]. In these cohorts, NAT has been associated with higher rates of margin-negative (R0) resection—often exceeding 80% compared with around two-thirds in surgery-first patients—and a marked reduction in pathological lymph node involvement [21,22]. Importantly, these improvements have been achieved without an increase in perioperative mortality, suggesting that NAT does not compromise short-term surgical safety when delivered in experienced centers [7]. However, such real-world datasets also reveal a striking degree of heterogeneity in how NAT is applied. Regimen selection varies widely, ranging from intensive multi-agent chemotherapy to gemcitabine-based combinations [23]; treatment duration can differ by several weeks; radiotherapy is used selectively, often without a unified rationale [24]; and the timing between completion of therapy and surgery is far from consistent [25]. This variability underscores the current lack of standardized protocols and complicates cross-center comparisons. Furthermore, survival comparisons between NAT and upfront surgery in retrospective or observational cohorts are inherently subject to immortal time bias (ITB), a statistical distortion that can overestimate the benefit of neoadjuvant strategies.

For clinicians, these findings carry a dual message. On one hand, they validate the oncologic and perioperative benefits of NAT outside of clinical trials [18]. On the other hand, they highlight the urgent need for harmonized treatment pathways, ideally grounded in consensus guidelines but adaptable to individual patient and tumor biology, to ensure that NAT’s potential is realized consistently across diverse practice settings [10,26].

## 6. Limitations and Remaining Controversies

Despite encouraging data from both randomized trials and high-quality observational studies, the implementation of NAT in PDAC continues to face significant conceptual and practical hurdles.

A fundamental limitation lies in the current anatomical definition of resectability, which does not account for tumor biology. Clinical experience and population studies have shown that some anatomically “resectable” PDACs recur within months of surgery despite complete (R0) resection, while certain borderline resectable cases achieve durable survival after NAT and surgery [27]. This discrepancy suggests that the anatomical classification, largely based on vascular involvement, is an incomplete surrogate for true biological resectability.

Equally problematic is the lack of robust predictive biomarkers. Carbohydrate antigen (CA) 19-9 remains the most widely used marker, and its weaknesses are inherently linked to the marker itself: poor specificity, elevation in benign conditions, and lack of expression in approximately 10% of patients due to Lewis antigen-negative phenotype. Nevertheless, CA19-9 retains clinical value when interpreted in the appropriate context, supporting both preoperative risk stratification and monitoring of treatment response during NAT. Recent evidence has further expanded its potential utility: Newhook et al. demonstrated that CA19-9 normalization during NAT was independently associated with improved survival, and that distinct dynamic trajectories of CA19-9 levels provided even more refined prognostic stratification than normalization alone. These findings indicate that, despite intrinsic shortcomings, CA19-9 remains a relevant and context-dependent tool in the neoadjuvant setting [28]. More sophisticated tools such as circulating tumor DNA (ctDNA) analysis and radiomics-based imaging signatures have shown promise in early studies for predicting response to NAT and long-term outcomes [29,30], but neither has yet been validated in prospective, multicenter trials.

Assessment of treatment response poses another challenge. Radiographic evaluation after NAT frequently underestimates residual tumor due to therapy-induced desmoplastic fibrosis, making it difficult to distinguish true response from non-viable tissue [31]. Similarly, pathologic complete responses (pCR) remain rare in PDAC, occurring in fewer than 5% of resected cases even after intensive multi-agent NAT [32]. This limits the use of pCR as a reliable surrogate endpoint, in contrast to other malignancies such as breast or rectal cancer.

Treatment heterogeneity, in the choice of chemotherapy backbone, total duration of preoperative treatment, and selective or routine inclusion of radiotherapy, further complicates evidence synthesis. In real-world series, regimens range from short-course gemcitabine-based therapy to prolonged multi-agent protocols such as mFOLFIRINOX, with or without chemoradiotherapy, and there is no consensus on optimal sequencing [23,24].

Finally, the risk of progression during NAT, although relatively uncommon in experienced centers, remains a key psychological barrier to wider adoption, particularly among surgeons accustomed to a surgery-first approach. Importantly, data suggest that progression during NAT usually reflects aggressive tumor biology rather than therapy-induced delay [7], but the perception of “missing the surgical window” continues to influence decision-making in multidisciplinary settings.

Taken together, these limitations highlight the need for a paradigm shift from purely anatomical selection criteria toward a multifactorial, biology-informed framework, integrating molecular, serological, and radiological data to guide NAT decisions and assess treatment benefit. Until such precision tools are available and validated, variability in practice and uncertainty in patient selection will continue to temper the enthusiasm for universal adoption of NAT in all resectable PDAC cases.

## 7. Future Perspectives: From Anatomy to Biology-Driven NAT

Looking ahead, the most transformative shift will come from advances in systemic therapy and deeper understanding of PDAC biology, such as the development of effective targeted strategies and identification of resistance mechanisms. Radiographic criteria will therefore remain essential for defining surgical resectability and planning, while biological and biochemical insights will complement rather than replace imaging in refining patient selection and treatment sequencing. Molecular profiling has already revealed distinct transcriptional subtypes of PDAC, such as classical and basal-like, with differing prognoses and [33]. Integrating this information into NAT decision-making could allow clinicians to match regimens to tumor biology, improving response rates and long-term outcomes.

Liquid biopsy technologies, particularly ctDNA analysis, offer an additional dimension by enabling real-time monitoring of tumor burden and treatment effect. Dynamic changes in ctDNA during NAT could serve as early predictors of response or progression, guiding escalation, de-escalation, or a switch in therapy before surgery. Likewise, advanced imaging modalities, including functional MRI and PET/MRI, may help distinguish viable tumor from post-treatment fibrosis, improving surgical planning and potentially avoiding non-beneficial resections.

Trial methodology is also evolving. Adaptive trial designs that incorporate early biomarker read-outs, along with biomarker-enriched basket trials, are already underway (e.g., NCT03983057, NCT03322995). These approaches allow for treatment modification in real time and for testing targeted agents in molecularly selected populations. Promising avenues include immunotherapy combinations in tumors with immunogenic phenotypes, targeted therapy in biomarker-defined subsets such as BRCA-mutated PDAC, and perioperative strategies tailored to genomic and transcriptomic profiles.

While these innovations are still in early phases, their convergence with NAT has the potential to redefine the treatment pathway for PDAC, moving from a “one-size-fits-all” approach to a precision, response-adapted strategy that maximizes benefit while avoiding unnecessary toxicity or delay.

## 8. Conclusions

NAT has progressed from a controversial, selectively applied option to the standard of care for borderline resectable PDAC, and it is now being increasingly considered for anatomically resectable cases with high-risk or biologically aggressive features. This evolution has been driven by the convergence of guideline recommendations, supportive evidence from randomized controlled trials, and corroborating real-world data from high-volume centers. Collectively, these data support the view that NAT may improve rates of R0 resection, reduce lymph node positivity, and maintain perioperative safety, while offering the unique advantage of delivering systemic therapy to all eligible patients prior to surgery.

The challenge now is to refine NAT into a precision oncology tool rather than a uniform preoperative step. This will require integrating biological markers, including genomic and transcriptomic profiles, ctDNA kinetics, and advanced imaging signatures, into patient selection and treatment adaptation. The shift from anatomy-only to anatomy-plus-biology selection will enable more accurate identification of patients most likely to benefit, while sparing others the toxicity and potential delays associated with ineffective therapy.

Standardization of regimens, treatment duration, and the role of radiotherapy will also be essential to reduce the heterogeneity currently observed in real-world practice and to facilitate meaningful comparison across centers and trials. As illustrated by the active clinical trials summarized in Table 2, current research is exploring multiple strategies, from optimized chemotherapy combinations to integrated chemo-immunotherapy and multimodal approaches, across both resectable and borderline resectable disease. These studies not only reflect the rapid diversification of NAT but also provide the framework for evidence-based refinement of patient selection and treatment sequencing.

At the same time, adaptive trial designs and biomarker-enriched studies will accelerate the translation of emerging targeted and immunotherapeutic strategies into the neoadjuvant space. As these elements coalesce, NAT has the potential to move beyond its current role as a bridge to surgery. It could become a central platform for precision oncology in PDAC—one that not only optimizes surgical outcomes but also serves as an in vivo testbed for systemic therapy efficacy, informs postoperative treatment strategies, and ultimately contributes to improved long-term survival in this highly lethal disease.

## Figures and Tables

**Table 1 cancers-17-03085-t001:** Summary of key trials evaluating NAT in resectable and borderline PDAC.

Study (Year)	Population	NAT Regimen	Control	Primary Endpoint	Main Results
PREOPANC-1 (2020)	Resectable (54%), borderline (46%)	Gem + RT (36 Gy) → surgery → Gem	Surgery → Gem	OS	OS 15.7 vs. 14.3 mo; ↑ R0, ↑ DFS
Prep-02/JSAP-05 (2019)	Resectable (80%), borderline (20%)	Gem + S-1 → surgery → S-1	Surgery → S-1	OS	OS 36.7 vs. 26.6 mo (HR 0.72)
ESPAC-5F (2020)	Borderline	FOLFIRINOX or GemCap ± RT	Upfront surgery	1-year OS	77% vs. 39% (HR 0.29)
SWOG S1505 (2021)	Resectable	mFOLFIRINOX vs. Gem/nab-P	–	OS	~23 mo both arms
NORPACT-1 (2023)	Resectable	mFOLFIRINOX (2 mo) → surgery	Upfront surgery	OS	No OS benefit
Alliance A021101 (2016)	Borderline	mFOLFIRINOX → Cap-RT	–	Safety/Efficacy	Resection 68%, R0 93%

**Table 2 cancers-17-03085-t002:** Active trials (neoadjuvant therapy for resectable/borderline resectable PDAC).

NCT	Short Title	Population	Design/Regimens	Phase	Status
NCT04927780 (PREOPANC-3)	Peri-op mFOLFIRINOX vs. adjuvant mFOLFIRINOX	Resectable	Randomized: 8 cycles neo mFOLFIRINOX → surgery → 4 cycles adj vs. surgery → 12 cycles adj	III	Active (ongoing)
NCT06210360	Perioperative vs adjuvant chemotherapy in high-risk resectable	High-risk resectable	Randomized: neo CT + surgery + adj CT vs. surgery + adj CT	III	Recruiting
NCT06423326	Gemcitabine + Cisplatin + nab-paclitaxel neoadjuvant	Resectable/BR	Single arm: Gem/Cis/nab-P before surgery	II	Recruiting
NCT05562297	Sintilimab + Gem/nab-paclitaxel neoadjuvant	Resectable/BR	Chemo-immunotherapy combination pre-op	II	Active (ongoing)
NCT06384560	Triple: chemo + immuno + RT neoadjuvant	Borderline resectable	Integrated pre-op scheme → surgery	II	Recruiting
NCT04617821	AG (Gem/nab-P) vs. mFOLFIRINOX neoadjuvant	BR (includes LA)	Randomized 1:1	III	Recruiting (includes BR; note: also includes LA)
NCT05679050	Sequential AG → FOLFIRINOX neoadjuvant	Resectable/BR	Single arm sequential pre-op	II	Recruiting
NCT06816914	NALIRIFOX perioperative	Resectable	Safety/activity peri-op study	II	Recruiting
NCT07034703 (PANACHE02-screening)	Prospective cohort treated with neoadjuvant	Resectable	Observational/prospective neo-pathway	—	Recruiting

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
