# Peer review of "Neoadjuvant Therapy in Pancreatic Ductal Adenocarcinoma: Aligning Guideline Recommendations with Real-World Evidence"

_cancers, 2025, doi:10.3390/cancers17183085_

Round 1

Reviewer 1 Report

Comments and Suggestions for Authors

Thank you for the opportunity to review this manuscript.  As a surgical oncologist specializing in PDAC, I find this manuscript impactful and timely.  The manuscript is well written. The manuscript reviews a hot topic;  a topic debated still at international meetings and within institutions and between providers and tumor boards.  Importantly, we have to use the art of medicine to provide the best patient outcome for those with PDAC in spite of underwhelming evidence and underwhelming technologies, and underwhelming systemic therapies.  It's an imperfect science.  I have a few comments that should augment this manuscript review and ensure it is up to date considering recently pubs.

1) in abstract line 35, this is a bit too vaulted of a statement regarding NAT and precision oncology considering we only have multi drug cytotoxic chemo currently.  Until we actually have targeted therapies in the surgical patient should we be talking about precision oncology......  Perhaps just removing precision oncology platform from the abstract.

2) reference 3 should be updated to the 2025 publication and the 2025 statistics in PDAC should be used.

3) information in line 73-75 should also reference Rompen IF, et al 2025 in Surgery regarding margin status.

4) lines 84-86 should have a reference.  suggest Versteijne E PREOPANC trial results in JCO and recent others.

5) lines 87-92 I agree with however I think you all should add information about the risks/adverse events related to NAT and reference those.  NAT is not easy on many patients and some don't get to surgery because of complications related to NAT and biliary interventions.  To increase the comprehensive nature of this manuscript these naunces should be acknowledged.

6) in section 4, I think it is important to acknowledge that currently (with underwhelming systemic therapies) surgery is the "most bang for your buck" or has the best efficacy as opposed to systemic therapy.  While multimodality therapy is ideal.....the patient usually should get surgery if possible (meaning resectable).  So, resectability criteria is for surgery and surgeons and should not be confused with precision medicine until we have better sytemic therapies or understanding of who should not get surgery due to biologic concerns.  With this (arguable) perspective in mind, line 129 may not be true.  The manuscript should provide and reference evidence/data showing the rate of surgery in those who receive NAT.  Also, the manuscript should reference and provide data regarding who many patients actually receive the planned multimodality therapy (in the real world or in clinical trials). Surgeons remain worried that biliary complications and NAT complications will prevent curative intent surgery which remains the "most bang for your buck".

7) a recent review regarding NAT trials in resectable disease has been published: Cardell CF et al Annals of Surgical Oncology 2025.  The narrative should align with this recent publication including any updated evidence.

8) CA 19-9 is an important tumor marker and evidence supports it as a good predictive marker.  Please review and add to manuscript: Newhook TE et al Annals of Surgery 2023

9) in line 228-230; I respectfully disagree.  radiographic criteria is designed for surgery which is the best treatment we currently have for locoregional control.  Not until we have better systemic therapies will we be able to "evolve" past current radiographic criteria of locoregional disease vs. systemic disease.  so seems to me the most transformation shift will be improved understanding of PDAC biology and biochemistry (ie RAS inhibitor development) and better understanding of resistance targets.  We know PDAC is a systemic disease at all stages.  Therefore, to impact the systemic disease we got to do better at treating that.  and that will be transformative in this disease.

Author Response

We thank the reviewer for the thoughtful and constructive comments, which have greatly improved the clarity and balance of our manuscript. Below we provide a point-by-point response. All suggested changes have been incorporated into the revised version of the manuscript.

Reviewer 1 – Comment 1

In abstract line 35, the statement regarding NAT as a precision oncology platform is too strong. Until targeted therapies are available, it may be premature.

Response 1

We thank the reviewer for this important observation. We agree that the concept of NAT as a “precision oncology platform” may appear overstated at this stage, given that current regimens are still based on multi-drug cytotoxic chemotherapy. We have therefore modified the abstract, removing the reference to a “precision oncology platform” and instead referring to NAT as an evolving strategic approach to optimize multimodal treatment delivery and patient selection.

Reviewer 1 – Comment 2

Reference 3 should be updated to the 2025 publication and the 2025 statistics in PDAC should be used.

Response 2

We appreciate this suggestion. We have updated reference 3 to the most recent CA Cancer J Clin 2025 statistics, ensuring that the epidemiological data cited in the Introduction reflects the latest available estimates.

Reviewer 1 – Comment 3

Information in lines 73–75 should also reference Rompen IF, et al 2025 in Surgery regarding margin status.

Response 3

Thank you for highlighting this recent publication. We have incorporated the reference Rompen IF, et al. Surgery. 2025 into the discussion of margin status, thereby strengthening our statement regarding the prognostic importance of R0 resection after NAT.

Reviewer 1 – Comment 4

Lines 84–86 should have a reference. Suggest Versteijne E PREOPANC trial results in JCO and recent others.

Response 4

We fully agree. We have now added Versteijne E, et al. J Clin Oncol. 2020 (PREOPANC trial) to support this statement, along with other recent references, to ensure that this section is well substantiated.

Reviewer 1 – Comment 5

Add information about the risks/adverse events related to NAT (some patients never reach surgery due to toxicity or biliary complications).

Response 5

We thank the reviewer for this valuable suggestion. In line with the comment, we have expanded the manuscript to acknowledge that NAT is not without risks. In particular, chemotherapy-related toxicities and biliary stent–associated morbidity can prevent a proportion of patients from reaching surgery. Prospective data indicate that approximately 10–20% of patients may fail to undergo resection due to treatment-related adverse events or progression during NAT [Murphy JE, et al. JAMA Oncol. 2018]. We believe that including these nuances provides a more balanced view of both the potential benefits and the limitations of NAT.

Reviewer 1 – Comment 6

In section 4, acknowledge that surgery remains the treatment with the “most bang for your buck.” Provide data on the proportion of patients actually receiving surgery and completing multimodality therapy after NAT.

Response 6

We thank the reviewer for this important point. We have revised Section 4 to emphasize that, despite the promise of multimodality therapy, surgery remains the most eligible treatment option for patients with resectable PDAC. To address the reviewer’s request, we added a new paragraph highlighting attrition before surgery as a key limitation of the neoadjuvant approach. Specifically, in randomized trials, only 60–75% of patients ultimately underwent resection (61% in PREOPANC-1 [14], ~77% in SWOG S1505 [18], and 60% in NORPACT-1 [19]), and less than half completed the entire multimodality pathway including postoperative therapy. These findings were integrated into Section 4 to provide a balanced perspective, acknowledging that while NAT offers relevant oncologic advantages, curative-intent surgery remains the most eligible and decisive therapeutic step in the management of resectable PDAC.

Reviewer 1 – Comment 7

A recent review regarding NAT trials in resectable disease has been published: Cardell CF et al, Annals of Surgical Oncology 2025. Align with this.

Response 7

We thank the reviewer for highlighting this important review. We have now incorporated Cardell CF, et al. Ann Surg Oncol. 2025 into the revised manuscript and aligned our discussion of NAT in resectable PDAC with the evidence summarized therein. In particular, we referenced this work to underscore that, while multiple trials confirm the feasibility of neoadjuvant strategies in anatomically resectable patients, the survival benefit remains heterogeneous and less consistent than in borderline resectable disease.

Reviewer 1 – Comment 8

CA19-9 is important and evidence supports it as predictive. Please review and add Newhook TE et al, Annals of Surgery 2023.

Response 8

We appreciate this observation. In the revised manuscript, we have clarified that the limitations of CA19-9 are inherent to the marker itself—namely poor specificity, frequent elevation in benign biliary conditions, and lack of expression in approximately 10% of patients due to Lewis antigen–negative phenotype. Nevertheless, we emphasized that CA19-9 retains clinical value when interpreted in the appropriate context, both for preoperative risk stratification and for monitoring response during NAT. Importantly, we incorporated the recent study by Newhook TE, et al. Ann Surg. 2023, which demonstrated that not only normalization but also distinct dynamic trajectories of CA19-9 during NAT are independently associated with survival after resection, providing more refined prognostic information than normalization alone. These additions strengthen the discussion by acknowledging both the intrinsic limitations and the context-dependent utility of CA19-9 in the neoadjuvant setting.

Reviewer 1 – Comment 9

In line 228–230; I respectfully disagree. Radiographic criteria is designed for surgery which is the best treatment we currently have for locoregional control. Not until we have better systemic therapies will we be able to “evolve” past current radiographic criteria of locoregional disease vs. systemic disease. So seems to me the most transformative shift will be improved understanding of PDAC biology and biochemistry (ie RAS inhibitor development) and better understanding of resistance targets.

Response 9

We thank the reviewer for this thoughtful perspective. We have revised the relevant section of the manuscript to clarify that radiographic criteria remain essential for surgical decision-making and for defining resectability, given that surgery is still the most effective treatment for locoregional control. We agree that, until more effective systemic therapies are available, these criteria cannot be replaced. In the revised text, we emphasized that the most transformative advances are likely to come from a deeper understanding of PDAC biology and resistance mechanisms (e.g., the development of RAS-targeted therapies) combined with improvements in systemic treatment efficacy. Radiographic criteria will therefore continue to serve as the cornerstone for surgical planning, while biological and biochemical insights will complement them by helping to refine patient selection and guide treatment sequencing.

Reviewer 2 Report

Comments and Suggestions for Authors

This is a well-written study on one of the most active areas in surgical oncology.

  1. The abstract is currently dense, with excessive trial details. Consider condensing and emphasizing the central perspective argument rather than listing multiple trial outcomes.
  2. While the language is grammatically sound, the manuscript reads more certain than it should for a perspective piece on an evolving field.
  3. The paper accurately summarizes trial results, but could better contextualize effect sizes
  4. In real-world series and retrospective datasets, survival comparisons between NAT and upfront surgery are at risk of immortal time bias (ITB). Patients in the NAT arm must survive long enough to complete preoperative therapy and reach surgery, whereas surgery-first patients are “at risk” from diagnosis. Discussing this limitation would strengthen the manuscript’s statistical rigor.
  5. References are overall appropriate, though some citations could be updated with more recent studies to increase timeliness.

Author Response

Response to Reviewer 2

We thank the reviewer for the thoughtful and constructive comments, which have greatly improved the clarity and balance of our manuscript. Below we provide a point-by-point response. All suggested changes have been incorporated into the revised version of the manuscript.

Reviewer 2 – Comment 1

The abstract is currently dense, with excessive trial details. Consider condensing and emphasizing the central perspective argument rather than listing multiple trial outcomes.

Response 1

We agree that the current abstract is quite detailed regarding trial outcomes. We have revised it to provide a more concise overview that emphasizes the central perspective and thematic argument, rather than enumerating multiple study results.

Reviewer 2 – Comment 2

While the language is grammatically sound, the manuscript reads more certain than it should for a perspective piece on an evolving field.

Response 2

We thank the reviewer for this important observation. We have adjusted the wording to better reflect the ongoing uncertainties and the evolving nature of evidence in neoadjuvant therapy for PDAC.

Reviewer 2 – Comment 3

The paper accurately summarizes trial results, but could better contextualize effect sizes

Response 3

We have expanded the discussion to better contextualize effect sizes, emphasizing their clinical relevance and limitations to provide a more balanced interpretation of the findings.

Reviewer 2 – Comment 4

In real-world series and retrospective datasets, survival comparisons between NAT and upfront surgery are at risk of immortal time bias (ITB). Patients in the NAT arm must survive long enough to complete preoperative therapy and reach surgery, whereas surgery-first patients are “at risk” from diagnosis. Discussing this limitation would strengthen the manuscript’s statistical rigor.

Response 4

We appreciate the suggestion to explicitly discuss the risk of immortal time bias (ITB) inherent in real-world and retrospective survival comparisons between NAT and upfront surgery. We have added a dedicated section addressing this statistical limitation to strengthen the rigor of the analysis.

Reviewer 2 – Comment 5

References are overall appropriate, though some citations could be updated with more recent studies to increase timeliness.

Response 5

We have reviewed and updated the reference list to include the most recent relevant studies, ensuring that the manuscript reflects the latest evidence and maintains timeliness.
